# Perceived ease of use of Truenat among laboratory staff as a new diagnostic tool for TB in Nigeria: Result of a pilot roll-out

Austin Ihesie[1], Nkiru Nwokoye[2], Jamiu Olabamiji[3], Kingsley Ochei[4], Rupert Eneogu[1], Debby Nongo[1], Aminu Babayi[2], Rayi Olugbemiga[3], Bethrand Odume[2], Aderonke Agbaje[3], Omosalewa Oyelaran[1], Wayne van Germert[5], Lucy Mupfumi[5], Omoniyi Amos[6], Elom Emeka[7], Chukwuma Anyaike[7], Edmund Ndudi Ossai[8]*

1 HIV AIDS & TB Office, USAID Nigeria, Abuja, Nigeria, 2 KNCV Nigeria, Abuja, Nigeria, 3 IHV-Nigeria, Lagos, Nigeria, 4 USAID Leap Project, Abuja, Nigeria, 5 STOP TB Partnership, Geneva, Switzerland, 6 World Health Organization, Abuja, Nigeria, 7 Federal Ministry of Health, Abuja, Nigeria, 8 Department of Community Medicine, College of Health Sciences, Ebonyi State University Abakaliki, Abakaliki, Nigeria

* ossai_2@yahoo.co.uk

**Data Availability Statement:** The datasets used and/or analyzed during this study are within the paper and its Supporting Information files.

## Abstract

In January 2020, WHO released a rapid communication on use of molecular assays as initial tests for diagnosis of tuberculosis, recommending Truenat as a replacement for smear microscopy in TB diagnosis. This study was designed to assess perceived ease of use of Truenat among Laboratory staff as a new diagnostic tool for TB in Nigeria. This study used a cross-sectional design. All trained Laboratory personnel operating the Truenat Duo equipment in 38 Truenat sites in the country were included. Information was obtained using a pre-tested self-administered questionnaire. Ease of use of Truenat was assessed using twenty-three variables on a five-point Likert scale of 1–5. The variables were analyzed quantitatively and qualitatively. Good Ease of use of Truenat was determined by proportion of respondents who answered $\geq$70% of the variables in the positive. All 50 Truenat Laboratory staff participated in the study. (Response rate 100%). Majority, (58%) were male. The median estimated number of tests before Laboratory staff became proficient with Truenat machine was 9 (IQR, 4–20), median number of tests to be analyzed within eight working hours was 10, (IQR = 8–15) and median time to conduct a Truelab MTBPlus test from start to finish was 60 minutes (IQR = 60–80). The commonest operational challenge that required Molbio service support was Trueprep errors/blockage, 47.4%. Overall, mean ease of use score was 4.0±0.4. Majority, (76%) had Good Ease of use of Truenat. No factor significantly influenced Ease of use of Truenat. Truenat machine is easy to use for a trained laboratory staff with minimal technical support and hence could be rolled out easily and successfully by various National TB Programs. Considering the high Trueprep challenges reported, there is need for further studies into the common errors/challenges, the contexts surrounding them and the programmatic intervention to address the high rate of Trueprep equipment faults.

**Funding:** The study was funded by the United States Agency for International Development (USAID) through the Stop TB partnership introducing new tools project (iNTP). The USAID Country Team in Nigeria was involved in the design and implementation of the study and in producing the manuscript. However, the views as expressed in the manuscript do not represent that of USAID.

**Competing interests:** The authors have declared that no competing interest exist.

## Introduction

Globally, an estimated 10.6 million people fell in with tuberculosis (TB) in the year 2021. This reveals an increase of 4.5% over the 2020 estimate of 10.1 million people. Additionally, the burden of drug-resistant TB (DR-TB) also increased within the same period with the diagnosis of 450,000 new cases of Rifampicin resistant TB in 2021 [1]. Similarly, the number of deaths associated with tuberculosis increased from an estimated 1.4 million deaths in 2020 to 1.6 million in 2021 [1]. The COVID-19 pandemic resulted in a large global fall in the number of people newly diagnosed with TB and notified in the year 2020 compared with the figures of 2019. Suffice it to say that COVID-19 pandemic had a negative effect on access to TB diagnosis and treatment and the burden of TB disease globally in 2021 [1].

In Nigeria, although the COVID-19 pandemic affected efficient TB program implementation, the country was among four countries worldwide that experienced an increase in TB case finding within the period of the COVID-19 pandemic [2]. Notwithstanding, TB diagnosis remains a challenge in Nigeria as the country accounts for 4.6% of the global TB burden. The country is also one of the ten countries that contributes to the highest burden of missing TB cases globally and also has a high triple burden of TB, DR-TB and HIV associated TB [3].

It has been posited that a rapid and widely available diagnostic test for TB with sensitivity of 85% for smear-positive and smear-negative cases, and a specificity of 97% can save 400,000 lives annually [4]. For many years AFB microscopy was the mainstay of TB diagnosis in Nigeria until 2010, when due to several limitations (including low sensitivity and inability to diagnose Rifampicin resistant TB), the molecular GeneXpert MTB/Rif assay was recommended by World Health Organization (WHO) to be the initial diagnostic test. However, the challenges associated with infrastructural and environmental requirements for successful GeneXpert implementation especially in resource challenged settings like Nigeria have resulted in limitations in coverage and accessibility. Despite having 507 installed GeneXpert machines, the proportion of bacteriologically diagnosed TB patients notified by the country in 2021 who were tested with molecular WHO-recommended Diagnostic tests (mWRDs) was just 79% [1]. The limited access to mWRDs also reflected in the low DR-TB notification for the country, implying a pool of missing DR-TB patients yet to be diagnosed and placed on treatment.

In January 2020, WHO released a rapid communication on the use of molecular assays as initial tests for the diagnosis of tuberculosis, recommending Truenat as a replacement for smear microscopy in TB diagnosis & detection of rifampicin resistance [5]. Truenat Test is a rapid molecular test manufactured by Molbio, an Indian Company, for diagnosis of active TB. It is a portable, battery-operated, chip-based test which detects Mycobacterium tuberculosis (MTB) in approximately one hour and rifampicin (RIF) resistance in another 40–60 minutes. It can be used at ambient temperatures up to 40 degrees Celsius. Truenat presents as a cost-effective molecular test that can be used as a near point-of-care test in peripheral facilities and "low-infrastructure" settings to diagnose TB and potentially save thousands of lives [6].

Truenat assays have a high sensitivity (80–87%) and specificity (95–97%) when compared with a composite reference standard consisting of GeneXpert and culture results [6]. There is also a high correlation (92.7%) between Xpert MTB/RIF and Truenat in MTB detection.[4] The chip-based test has been designed to simplify the process of real-time polymerase chain reaction (PCR) from 'sample to result' so that laboratories with minimal infrastructure can easily perform these tests routinely in their facilities and report PCR results in less than an hour [7].

Based on the recent WHO recommendation of Truenat, the Stop TB Partnership in 2021 procured and donated 38 Truenat Machines to Nigeria as part of the USAID-funded Introducing New Tools Project (iNTP). The iNTP project is the largest multi-country implementation of Truenat instruments. The project aimed to provide access to high quality, innovative

diagnostic tools for TB among populations in hard-to-reach areas. This study sought to determine the perceived ease of use of Truenat as a diagnostic tool for TB among Truenat Laboratory staff in Nigeria.

## Materials and methods

### Study setting

Nigeria is the most populated country in Africa and occupies an area of land which is approximately 923,768 square kilometers. Administratively, the country is divided into 36 states and the Federal Capital Territory. The states serve as the second tier of government. The states are also grouped into six geo-political zones including northwest, northeast and north-central zones. Others include the southwest, southeast and south-south geo-political zones. There are 774 local government areas which form the third tier of government.

This study was conducted in 14 states in Nigeria which were selected for the implementation of Truenat intervention and supported by the USAID-funded TB LON projects. (See Fig 1). The States included Kano, Katsina, Kaduna, Bauchi, Nasarawa and Taraba in the Northern part of the country as well as Akwa-Ibom, Anambra, Cross River, Delta, Rivers, Lagos, Osun and Oyo States located in the southern part of the country. The USAID TB-LON 3 project

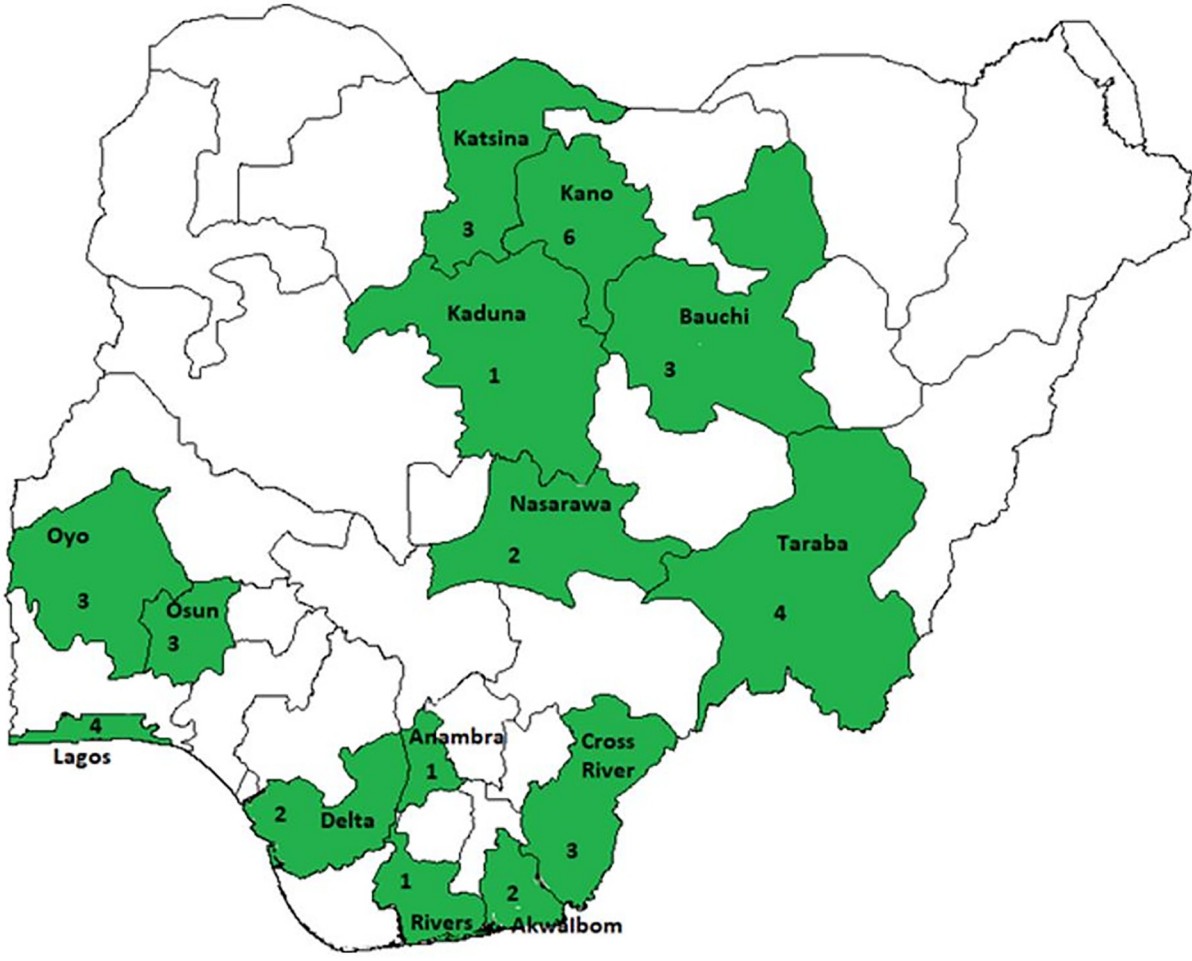

**Fig 1. Map of Nigeria indicating states implementing Truenat under the iNTP.**

implemented by the Institute of Human Virology of Nigeria (IHVN) is located in three states in the southwest geo-political zone of the country namely Lagos, Osun and Oyo states. The USAID TB-LON 1& 2 projects are located in the remaining 11 states implemented by KNCV Tuberculosis Foundation Nigeria.

## Study design

A health facility based cross-sectional study design was used.

## Study population

All trained Laboratory personnel operating the Truenat equipment in the 38 Truenat sites in the country who were willing to participate in the study were included.

## Methods and data collection

Data was collected using a pre-tested, semi-structured questionnaire. The questionnaire which was self-administered was designed by the researchers. Pre-testing of the questionnaire was conducted among eight Laboratory personnel who had been trained and operated the Truenat machine in the past. However, at the time of the study, these personnel were no longer working at the various health facilities where the Truenat machines were located.

## Study tools and variables

The questionnaire utilized both closed questions (including a five-point Likert scale numbered 1–5) and open-ended questions. The domains represented in the questionnaire included initiation of Truenat testing, sample preparation and testing, training, time required for Truenat testing, reporting, waste disposal, equipment breakdown and troubleshooting.

## Data management

Data entry and analysis were done using IBM Statistical Product for Service Solutions (SPSS) statistical software, version 25. Categorical variables were presented using frequencies and proportions while continuous variables were summarized using mean and standard deviation. Chi square test was used in the analysis and the level of statistical significance was determined by a $p$ value of $<0.05$.

The outcome measure of the study was Ease of use of Truenat, and this was assessed using twenty-three variables which elicited responses from all the respondents. The variables were measured using a five-point Likert scale where respondents provided ratings on a scale from 1–5. Specifically, 1 represented 'strongly disagree', 2 denoted 'disagree', 3 indicated 'neutral', 4 represented 'agree', while 5 represented 'strongly agree'. Reverse scoring was used for questions that were negatively worded. Each of the 23 variables was analyzed quantitatively and qualitatively using mean and standard deviation and frequencies and proportions. The mean and standard deviation for each of the 23 variables was obtained. Also, the mean of each of the six domains in the questionnaire including initiation of Truenat testing, sample preparation and testing, training, reporting, waste disposal, equipment breakdown and troubleshooting were also obtained. Then the overall mean score for the entire 23 variables used to assess ease of use was obtained.

Furthermore, for the purpose of analysis the responses to each of the variables were categorized into positive and negative (not represented in the tables). For each response in the positive, a score of one was given to each respondent while a score of zero was assigned for a negative response. The cumulative score for the 23 variables for each respondent was then obtained. Respondents who scored $\geq70\%$ of the total score were designated as having reported

'Good' Ease of use of Truenat, while respondents who scored <70% were regarded as having reported 'Poor' Ease of use of Truenat. The socio-demographic and facility-related characteristics of the respondents were then cross-tabulated with the outcome variable to determine factors associated with 'Good' Ease of use of Truenat and the level of statistical significance was determined by a $p$ value of <0.05.

### Ethical approval

Ethical approval for the study was obtained from the National Health Ethics Research Committee (NHREC). (Approval number NHREC/01/01/2007-05/04/2023). Participants were recruited for the study from April 11[th] to April 28[th] 2023 after which the process of data collection commenced. Each respondent signed a written informed consent form before participating in the study. The respondents were informed that participation in the study was voluntary, and they were free to withdraw from the study even after giving consent. The respondents were assured of anonymity and confidentiality of data.

### Results

Fifty Truenat Laboratory staff participated in the study representing a response rate of 100%. The mean age of the respondents was 36.5±7.9 years. A higher proportion, 58.0% were males. The majority of the respondents, 70.0%, were trained when the Truenat machines were installed. A higher proportion of the health facilities where the respondents worked, 86.0% were public facilities and also DOTS centers, 80.0% (Table 1).

Table 2 shows responses to variables for ease of initiation of testing and sample preparation. The mean score for being proficient with operating the Truenat machine was 4.5±0.5. The mean score for ease of initiation of Truenat testing was 4.6±0.4, while the mean score for ease of sample preparation for DNA extraction was 4.2±0.8. The mean score for ease of performing quality control procedures was 3.8±1.1. The mean score for the domain on sample preparation and testing was 3.8±0.5.

Table 3 shows the responses to variables for domains of training, breakdown and trouble-shooting. The mean score for the training domain was 4.3±0.5. The mean score for the reporting domain was 4.0±0.8. The mean score for waste disposal was 3.1±1.2. The mean score for breakdown and trouble-shooting domain was 4.0±0.7. The mean score for the training, breakdown/troubleshooting domain was 4.0±0.4.

Fig 2 shows responses for Truenat operations and proficiency. The median number of tests before the respondents got proficient with the Truenat machine was 9.0 tests. (IQR = 4.0–20.0). The median number of days needed by the respondents to become proficient with the Truenat machine was 4 days (IQR, 4-20days). The median number of tests the respondents are able to analyze in a normal eight working hours was 10 tests, (IQR = 8–15 days). The median time in minutes to conduct a Truelab MTBPlus test from start to finish was 60 minutes (IQR = 60–80 minutes). The median additional time in minutes to conduct the Truenat MTB-RIF test was 60 minutes (IQR = 60–80 minutes).

Table 4 shows the power source to charge the Truenat machine. The major sources of power supply for charging the Truenat machine included public power supply, 65.8% and generators, 65.8%. For respondents who reported insufficient training, areas highlighted for additional training included DNA extraction, 28.6% and full scale training, 28.6%.

Table 5 shows operational challenges of the Truenat machine. Commonest operational challenges that needed service support included Trueprep errors/blockage, 47.4% and connectivity/network issues, 13.2%. In 71.1% of cases, Trueprep has been repaired /replaced while for Truelab, it is 23.7%.

**Table 1. Socio-demographic characteristics of the respondents and characteristics of facilities.**

| Variable | Frequency (n = 50) | Percent (%) |
|---|---|---|
| **Age of respondents** | | |
| Mean±SD | 36.5±7.9 | |
| **Age of respondents in groups** | | |
| <30 years | 12 | 24.0 |
| 30–34 years | 9 | 18.0 |
| 35–39 years | 13 | 26.0 |
| ≥40 years | 16 | 32.0 |
| **Gender** | | |
| Male | 29 | 58.0 |
| Female | 21 | 42.0 |
| **No. of years of laboratory experience** | | |
| <10 years | 27 | 54.0 |
| ≥10 years | 23 | 46.0 |
| **No. of years in TB program** | | |
| <5 years | 20 | 40.0 |
| ≥5 years | 30 | 46.0 |
| **Respondent was part of initial set of Truenat users trained when machine was installed** | | |
| Yes | 35 | 70.0 |
| No | 15 | 30.0 |
| **Respondent type of health facility** | | |
| Public | 43 | 86.0 |
| Private | 7 | 14.0 |
| **Respondent facility is a DOTS center** | | |
| Yes | 40 | 80.0 |
| No | 10 | 20.0 |
| **Respondent facility is a microscopy center** | | |
| Yes | 30 | 60.0 |
| No | 20 | 40.0 |
| **Respondent facility is a center for TB LAMP** | | |
| Yes | 3 | 6.0 |
| No | 47 | 94.0 |

Fig 3 shows the perceived ease of use of Truenat among the respondents. A higher proportion of the respondents, 76.0% had Good Ease of use of Truenat.

Table 6 shows the factors associated with Good ease of use of Truenat among the respondents. A higher proportion of respondents who were 38 years and above(76.2%), had good ease of use of Truenat when compared with those who were less than 38 years old, 74.2% but the difference in proportions was not found to be statistically significant, ($\chi^2 = 0.146$, p = 0.702). Comparable proportions of male, 75.9% and female respondents, 85.7% had good ease of use of Truenat, ($\chi^2 = 0.001$,p = 0.979).

## Discussion

Findings from this study indicate that there is a high level of acceptability of Truenat by the end-users. In addition, there is a strong preference for using Truenat over microscopy among the Laboratory staff. From the results of our study, 70% of the respondents were part of those

**Table 2. Initiation of testing and sample preparations for Truenat.**

| Variable | Mean ±SD | Strongly disagree n (%) | Disagree n (%) | Neutral n (%) | Agree n(%) | Strongly Agree n(%) |
|---|---|---|---|---|---|---|
| **Initiation of Truenat testing in the center** | | | | | | |
| Truenat instrument system is easy to install and set up | 4.4 ±0.8 | 1 (2.3) | 0 (0.0) | 1 (2.3) | 20 (45.50 | 22 (50.0) |
| Initiating startup sequence steps for daily operations and handling of machine is easy | 4.6 ±0.5 | 0 (0.0) | 0 (0.0) | 0 (0.0) | 22 (44.0) | 28 (56.0) |
| Machine screen-display user interface is easy to operate for entering data | 4.6 ±0.5 | 0 (0.0) | 0 (0.0) | 0 (0.0) | 19 (38.0) | 31 (62.0) |
| I consider myself proficient with operating the Truenat machine | 4.5 ±0.5 | 0 (0.0) | 0 (0.0) | 1 (2.0) | 24 (48.0) | 25 (50.0) |
| **Initiation of Truenat testing (Average score)** | 4.6 ±0.4 | | | | | |
| **Sample preparation and testing** | | | | | | |
| Sample preparation for DNA extraction using Trueprep device is an easy operation | 4.2 ±0.8 | 1 (2.0) | 1 (2.0) | 5 (10.0) | 24 (48.0) | 19 (38.0) |
| Transfer of extracted DNA elute from elute collection tube to PCR reagents is an easy procedure | 4.5 ±0.5 | 0 (0.0) | 0 (0.0) | 1 (2.0) | 25 (50.0) | 24 (48.0) |
| Transfer of mixed DNA elute solution from microtube to MTBPlus chip reaction well is an easy procedure | 4.3 ±0.6 | 0 (0.0) | 0 (0.0) | 3 (6.0) | 29 (58.0) | 18 (36.0) |
| Further testing of MTB positive sample for RIF resistance is an easy procedure | 4.2 ±0.9 | 1 (2.0) | 3 (6.0) | 0 (0.0) | 27 (54.0) | 19 (38.0) |
| Quality control procedures are easy to follow and execute | 3.8 ±1.1 | 1 (2.0) | 7 (14.0) | 6 (12.0) | 21 (42.0) | 15 (30.0) |
| Non-availability of power for charging is a major structural challenge to the operation of Truenat | 2.7 ±1.4 | 11 (22.0) | 19 (38.0) | 4 (8.0) | 7 (14.0) | 9 (18.0) |
| Operating Truenat machine requires less laboratory consumables than AFB microscopy | 2.5 ±1.3 | 14 (28.0) | 16 (32.0) | 6 (12.0) | 10 (20.0) | 4 (8.0) |
| Operating Truenat machine requires less manual steps than AFB microscopy | 3.1 ±1.6 | 14(28.0) | 6 (12.0) | 5 (10.0) | 13 (26.0) | 12 (24.0) |
| I prefer using Truenat to AFB microscopy | 4.7 ±0.7 | 0 (0.0) | 1 (2.0) | 2 (4.0) | 10 (20.0) | 37 (74.0) |
| **Sample preparation and testing (Average score)** | 3.9 ±0.5 | | | | | |

trained when the Truenat machines were installed. In a study in India, 80% of the respondents had participated in a training on Truenat testing [4]. This could be an indication that training of healthcare workers, especially Laboratory staff on use of Truenat is a pre-requisite before or during the implementation period. There could have been staff attrition on the part of the Laboratory staff of those initially trained, hence those not trained at installation received on-the-job step-down training. The study results also showed that >90% of the respondents (including those who were not trained at installation) indicated that the training received was adequate, implying that they perceived the step-down training received to be adequate.

The median number of tests before the Laboratory staff gets proficient with the Truenat machine was 9 tests. This is similar to findings from a study in India where the median number of tests before the respondents became comfortable in performing Truenat tests was found to be 10 [4]. Also, the median number of days to attain proficiency with the Truenat machine was four days. This short period of time to attain proficiency in use of Truenat which was achieved post-training is an indication that the Truenat machine is easy to use by a trained Laboratory staff with minimal technical support and hence can be rolled-out easily. A study in India had described the Truenat TB test as a simple method and implied that laboratory technicians with

**Table 3. Training, breakdown and troubleshooting.**

| Variable | Mean±SD | *SD | Disagree | Neutral | Agree | *SA |
|---|---|---|---|---|---|---|
| **Training** | | | | | | |
| Training received for Truenat operation was adequate and sufficient | 4.1±0.8 | 0 (0.0) | 3 (6.0) | 4 (8.0) | 26 (52.0) | 17 (34.0) |
| Quality of post-training technical support received is adequate | 4.2±0.9 | 1 (2.0) | 3 (6.0) | 2 (4.0) | 25 (50.0) | 19 (38.0) |
| Easy to provide step down training to new lab staff on how to operate the machine | 4.5±0.5 | 0 (0.0) | 0 (0.0) | 1 (2.0) | 23 (46.0) | 26 (52.0) |
| **Training (Average score)** | 4.3±0.5 | | | | | |
| **Reporting** | | | | | | |
| Result from Truenat test is easy to report | 4.7±0.4 | 0 (0.0) | 0 (0.0) | 0 (0.0) | 13 (26.0) | 37 (74.0) |
| Easy to sort the results and get summaries of the tests done from the machine | 3.7±1.2 | 2 (4.0) | 8 (16.0) | 8 (16.0) | 15 (30.0) | 17 934.0) |
| Summary results for all the tests done within a period can be exported | 3.6±1.4 | 5 (10.0) | 8 (16.0) | 7 (14.0) | 14 928.0) | 16 (32.0) |
| **Reporting (Average score)** | 4.0±0.8 | | | | | |
| **Waste disposal** | | | | | | |
| Waste generated from Truenat testing requires its own special lab disposal methods that are available | 3.1±1.2 | 5 (10.0) | 13 (26.0) | 8 (16.0) | 21 (42.0) | 3 (6.0) |
| **Breakdown and troubleshooting** | | | | | | |
| Conducting daily maintenance on Truenat machine is easy | 4.2±0.7 | 0 (0.0) | 2 (4.0) | 2 (4.0) | 31 (62.0) | 15 (30.0) |
| Flushing the system when there is blockage is an easy procedure | 4.2±1.0 | 0 (0.0) | 5 (10.0) | 5 (10.0) | 17 (34.0) | 23 (46.0) |
| Replacement of the slider glass on Truelab bays is easy | 4.1±1.0 | 1 (2.0) | 3 (6.0) | 7 (14.0) | 16 (32.0) | 23 (46.0) |
| Technical support by service engineers is easy to get during breakdown and for troubleshooting | 3.5±1.3 | 6 (12.0) | 6 (12.0) | 6 (12.0) | 23 (46.0) | 9 (18.0) |
| **Breakdown and trouble-shooting (Average score)** | 4.0±0.7 | | | | | |
| **GRAND TOTAL SCORE (Average)** | 4.0±0.4 | | | | | |

* SD—Strongly disagree, SA—Strongly agree

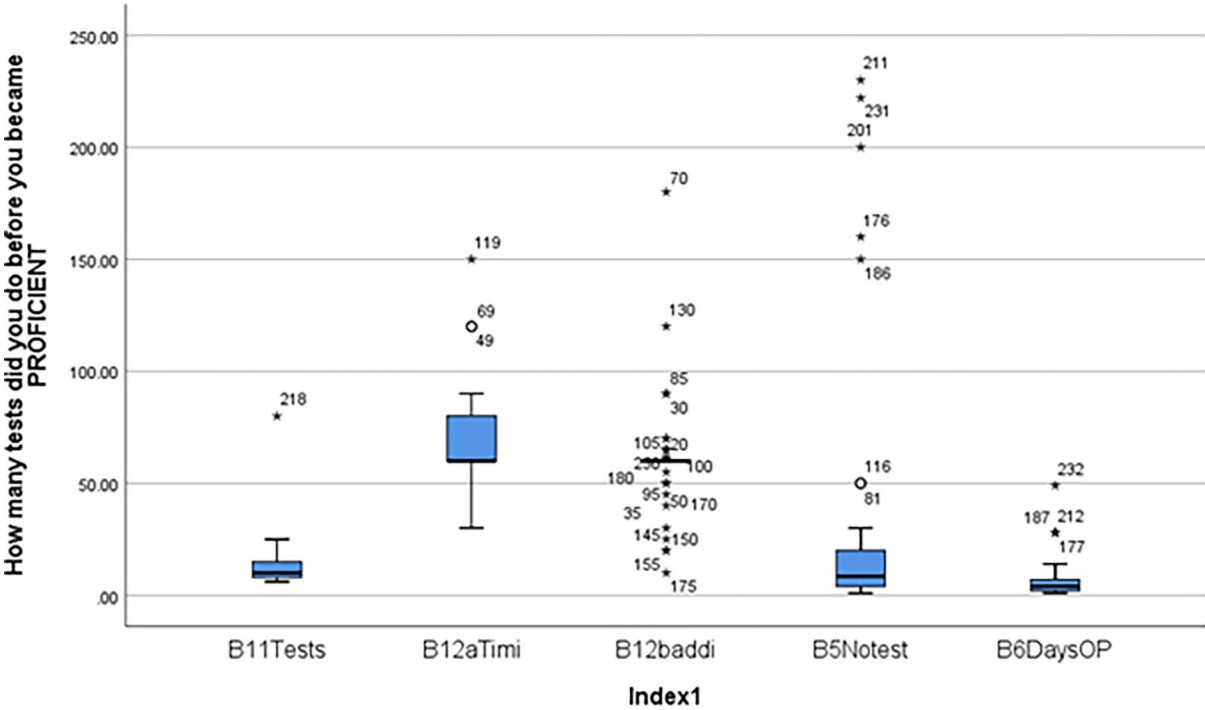

**Fig 2. Truenat operations and proficiency.** Key B11Tests; Estimated no of days before staff gets proficient with Truenat machine. B12aTimi: Estimated time in minutes to conduct a Truelab MTBPlus test from start to finish. B12baddi: Additional time in minutes to conduct Truenat MTB-RIF Dx test. B5Notest: No. of tests/samples that can be analyzed in an eight -hour shift. B6DaysOP: No. of days needed to become proficient with Truenat machine.

**Table 4. Truenat power source and additional training requirements.**

| Variable | Frequency (n = 38)** | Percent (%) |
|---|---|---|
| **Power source to charge Truenat machine*** | | |
| Public power supply | 25 | 65.8 |
| Generator | 25 | 65.8 |
| Solar | 6 | 15.8 |
| Inverter | 6 | 15.8 |
| None | 1 | 2.6 |
| **Where further training is required** | (n = 7) | |
| DNA extraction | 2 | 28.6 |
| Will require full training (received step down training) | 2 | 28.6 |
| Trueprep/Truelab | 1 | 14.3 |
| Maintenance | 1 | 14.3 |
| Superuser | 1 | 14.3 |

*multiple responses

** Number of Truenat sites

**Table 5. Operational challenges of Truenat machine.**

| Variable | Frequency (n = 38)* | Percent (%) |
|---|---|---|
| **Commonest operational challenge that needed Molbio service support** | | |
| Trueprep errors/blockage | 18 | 47.4 |
| Connection/network issues | 5 | 13.2 |
| Invalid result/error from Truelab | 4 | 10.5 |
| No challenge | 11 | 28.9 |
| **Sample rejected because of inability to test** | | |
| Yes | 30 | 78.9 |
| No | 8 | 21.1 |
| **Reason for rejection*** | (n = 30) | |
| Machine faulty | 19 | 63.3 |
| Sample insufficient/bloody | 7 | 23.3 |
| Reagent stock-out/expired | 3 | 10.0 |
| Power challenges | 1 | 3.3 |
| **Truelab replaced/repaired at any time** | (n = 38) | |
| Yes | 9 | 23.7 |
| No | 29 | 76.3 |
| If yes, how many times | (n = 9) | |
| Once | 4 | 44.4 |
| Twice | 5 | 55.6 |
| **Trueprep replaced/repaired at any time** | (n = 38) | |
| Yes | 27 | 71.1 |
| No | 11 | 28.9 |
| If yes, the number of times | (n = 27) | |
| Once | 10 | 37.0 |
| Twice | 7 | 26.0 |

(*Continued*)

**Table 5.** (Continued)

| Variable | Frequency (n = 38)* | Percent (%) |
|---|---|---|
| Three times and above | 10 | 37.0 |
| **Micro-printer replaced/repaired at any time** | (n = 38) | |
| Yes | 4 | 10.5 |
| No | 34 | 89.5 |
| If yes, the number of times | (n = 4) | |
| Once | 2 | 50.0 |
| Twice | 2 | 50.0 |

** Number of Truenat sites

minimal training can perform the test using the machine [8]. This also conforms to the result of this study in which the majority of respondents, 76% agreed or strongly agreed on the adequacy of the quality of training on use of Truenat received.

The median time in minutes to conduct a Truelab MTBPlus test from start to finish in this study was 60 minutes. A previous study also found that it took the Truenat MTB test about 60 minutes for the detection of TB [9]. Another study in India also concluded that the turn-around time for Truenat is approximately one hour [8]. The final report on operational feasibility and performance of Truenat MTB RIF assays in field settings in India concluded that the Truenat assay is designed to complete the detection of mycobacterium TB in 35 minutes [10]. From the results of this study, the median time to complete both MTB detection and RIF resistance was 120 minutes. An early study from India reported Truenat simultaneously detected RIF resistance and mycobacterium tuberculosis in less than 3 hours [11].

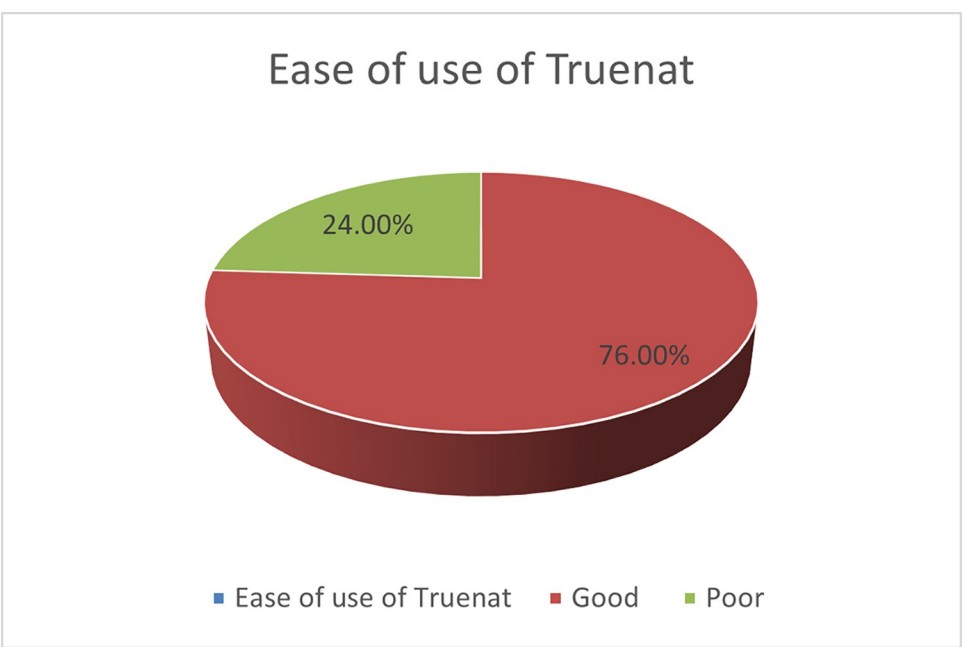

**Fig 3. Perceived ease of use of Truenat.**

**Table 6. Factors associated with good ease of use of Truenat among the respondents.**

| Variable | Ease of use of TRUENAT (n = 50) | | $\chi^2$ p value |
|---|---|---|---|
| | Good N (%) | Poor N (%) | |
| **Age of respondents** | | | |
| <38 years | 23(74.2) | 8 (25.8) | 0.146 0.702 |
| ≥38 years | 15 (78.9) | 4 (21.1) | |
| **Gender** | | | |
| Male | 22 (75.9) | 7 (24.1) | 0.001 0.979 |
| Female | 16 (76.2) | 5 (23.8) | |
| **Type of health facility** | | | |
| Public health facility | 31 (72.1) | 12 (27.9) | FT 0.174 |
| Private health facility | 7 (100.0) | 0 (0.0) | |
| **No of years of lab experience** | | | |
| <10 years | 20 (74.1) | 7 (25.9) | 0.119 0.730 |
| ≥10 years | 18 (78.3) | 5 (21.7) | |
| **No of years in National TB program** | | | |
| <5 years | 16 (80.0) | 4 (20.0) | 0.292 0.589 |
| ≥5 years | 22 (73.3) | 8 (26.7) | |
| **Facility is a DOTS center** | | | |
| Yes | 32 (80.0) | 8 (20.0) | 1.754 0.185 |
| No | 6 (60.0) | 4 (40.0) | |
| **Facility is a microscopy center** | | | |
| Yes | 23 (76.7) | 7 (23.3) | 0.018 0.892 |
| No | 15 (75.0) | 5 (25.0) | |
| **Facility is a TB LAMP center** | | | |
| Yes | 3 (100.0) | 0 (0.0) | 1.008 0.315 |
| No | 35(74.5) | 12 (25.5) | |
| **Received training when machine was installed** | | | |
| Yes | 27 (77.1) | 8 (22.9) | 0.084 0.773 |
| No | 11 (73.3) | 4 (26.7) | |

FT Fishers exact test

The commonest operational challenge that needed Molbio service support was Trueprep related problems. The study respondents reported three times more Trueprep challenges than they did for Truelab challenges with the Trueprep replaced at least once in 27 facilities compared to 9 facilities for the Truelab. The proportions in our study are higher than reported in a study in India involving 243 Truenat sites where seven sites had their Trueprep replaced at least once, while six sites had their Truelab device replaced at least once since installation [7]. With the increased use and scale up of Truenat to other tropical environments outside India it is expected that more information on operational performance, durability and challenges will become available.

Under sample testing almost all respondents agreed or strongly agreed that transfer of extracted DNA elute from elute collection tube to PCR reagents was an easy procedure. Same was the case for transfer of mixed DNA elute solution from micro-tube to MTBPlus chip reaction well. This is another confirmation that performing the Truenat test is easy for Laboratory staff who have received training on the use of Truenat. Also, 98% of the respondents agreed or strongly agreed that they consider themselves proficient in performing the Truenat test. Furthermore, 98% of the respondents agreed or strongly agreed that it is easy to provide step

down training to new Laboratory staff on how to operate the Truenat machine. Expectedly, the mean domain score for training was second only to initiation of Truenat testing in the facility. The average total score for all the variables was 4.0±0.4 which reveals a good ease of use of the Truenat machine by the respondents. A higher proportion of the respondents, 76.0% had good ease of use of Truenat. From the results of this study, no factor significantly influenced the good ease of use of the Truenat machine among the respondents.

## Conclusions

Truenat machine is easy to use for a trained laboratory staff with minimal technical support and hence could be rolled out easily and successfully by various National TB Programs. Considering the very high Trueprep challenges reported, there is need for further studies into the common errors/challenges, the contexts surrounding them and the programmatic intervention to address the high rate of Trueprep equipment faults.

## Supporting information

**S1 Data.**
(XLSX)

## Acknowledgments

The authors acknowledge the support by the National Tuberculosis and Leprosy Control Program, State TB Program Managers across the 18 USAID TB-LON supported states in Nigeria, Tuberculosis Local Government Supervisors, Health care workers in the selected sites, and study participants for their contribution to the success of this study. We also appreciate the technical support and input from the Stop TB partnership introducing new tools project (iNTP) team and USAID Washington TB Division.

## Author Contributions

**Conceptualization:** Austin Ihesie, Nkiru Nwokoye, Jamiu Olabamiji, Kingsley Ochei, Rupert Eneogu, Edmund Ndudi Ossai.

**Data curation:** Austin Ihesie, Nkiru Nwokoye, Jamiu Olabamiji, Kingsley Ochei, Edmund Ndudi Ossai.

**Formal analysis:** Edmund Ndudi Ossai.

**Funding acquisition:** Austin Ihesie, Rupert Eneogu.

**Investigation:** Austin Ihesie, Nkiru Nwokoye, Jamiu Olabamiji, Kingsley Ochei, Rupert Eneogu, Bethrand Odume, Aderonke Agbaje, Omosalewa Oyelaran, Lucy Mupfumi, Omoniyi Amos, Elom Emeka, Chukwuma Anyaike.

**Methodology:** Austin Ihesie, Nkiru Nwokoye, Jamiu Olabamiji, Kingsley Ochei, Edmund Ndudi Ossai.

**Project administration:** Austin Ihesie, Nkiru Nwokoye, Jamiu Olabamiji, Kingsley Ochei, Rupert Eneogu, Debby Nongo, Aminu Babayi, Rayi Olugbemiga, Bethrand Odume, Aderonke Agbaje, Omosalewa Oyelaran.

**Resources:** Austin Ihesie, Nkiru Nwokoye, Jamiu Olabamiji, Kingsley Ochei, Rupert Eneogu, Debby Nongo, Aminu Babayi, Rayi Olugbemiga, Bethrand Odume, Aderonke Agbaje,

Omosalewa Oyelaran, Wayne van Germert, Lucy Mupfumi, Omoniyi Amos, Elom Emeka, Chukwuma Anyaike.

**Software:** Debby Nongo, Aminu Babayi, Rayi Olugbemiga, Bethrand Odume, Aderonke Agbaje, Wayne van Germert, Lucy Mupfumi, Omoniyi Amos, Elom Emeka, Chukwuma Anyaike.

**Supervision:** Austin Ihesie, Nkiru Nwokoye, Jamiu Olabamiji, Kingsley Ochei, Rupert Eneogu, Debby Nongo, Aminu Babayi, Rayi Olugbemiga, Bethrand Odume, Aderonke Agbaje, Omosalewa Oyelaran, Wayne van Germert, Elom Emeka, Chukwuma Anyaike, Edmund Ndudi Ossai.

**Visualization:** Debby Nongo, Bethrand Odume, Wayne van Germert, Omoniyi Amos.

**Writing – original draft:** Austin Ihesie, Nkiru Nwokoye, Jamiu Olabamiji, Kingsley Ochei, Edmund Ndudi Ossai.

**Writing – review & editing:** Austin Ihesie, Nkiru Nwokoye, Jamiu Olabamiji, Kingsley Ochei, Rupert Eneogu, Debby Nongo, Aminu Babayi, Rayi Olugbemiga, Bethrand Odume, Aderonke Agbaje, Omosalewa Oyelaran, Wayne van Germert, Lucy Mupfumi, Omoniyi Amos, Elom Emeka, Chukwuma Anyaike, Edmund Ndudi Ossai.

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
