## [Decision Letter · Decision Letter 0]

17 Jul 2024

PGPH-D-24-01529

Perceived ease of use of Truenat among laboratory staff as a new diagnostic tool for TB in Nigeria: result of a pilot roll-out

Dear Dr. Ossai,

Thank you for submitting your manuscript to PLOS Global Public Health. After careful consideration, we feel that it has merit but does not fully meet PLOS Global Public Health’s publication criteria as it currently stands. Therefore, we invite you to submit a revised version of the manuscript that addresses the points raised during the review process.

We look forward to receiving your revised manuscript.

Kind regards,

Priya Rajendran, PhD

Academic Editor

Journal Requirements:

Additional Editor Comments (if provided):

Reviewers' comments:

Reviewer's Responses to Questions

**Comments to the Author**

1. Does this manuscript meet PLOS Global Public Health’s publication criteria? Is the manuscript technically sound, and do the data support the conclusions? The manuscript must describe methodologically and ethically rigorous research with conclusions that are appropriately drawn based on the data presented.

Reviewer #1: Yes

Reviewer #2: Yes

2. Has the statistical analysis been performed appropriately and rigorously?

Reviewer #1: Yes

Reviewer #2: Yes

3. Have the authors made all data underlying the findings in their manuscript fully available (please refer to the Data Availability Statement at the start of the manuscript PDF file)?

Reviewer #1: Yes

Reviewer #2: Yes

4. Is the manuscript presented in an intelligible fashion and written in standard English?

Reviewer #1: Yes

Reviewer #2: Yes

5. Review Comments to the Author

Reviewer #1: Line no. 226-228: Table 4 shows the power source to charge the Truenat machine. The major sources of power supply for charging the Truenat machine included public power supply, 65.8% and generators, 65.8%” Does this mean all the public power sources were using generators?

The two categories represented in Figure 3 are confusing. It would be better if the two responses were mentioned as “Good” and “Poor”

The study didn’t evaluate the performance and ease of use of Truenat in high burden settings. The average number of Truenat tests performed in the study centres was just 10, while most high-burden centres may receive 30 to 50 samples for TB diagnosis in a day. This aspect was not studied or included in the discussion.

The ease of use of Truenat has been compared only with microscopy. What was the opinion of the respondents in comparison to Genexpert?

Reviewer #2: The manuscript is well written and addresses one or the major challenges of performance of Truenat in programatic setting.

1. Truenat Rif testing has more indeterminate, has the authors try to capture as part of the performance.

2. Performance of Truenat compared to smear may be less labour intensive but compare to GeneXpert its more labour intensive and the chances of cross contamination during the testing may be high with out training. This has not been discussed.

3. The more common true prep issue of blockage is due to improper mucolysis or the machine problem has not been clearly marked in the manuscript.

6. PLOS authors have the option to publish the peer review history of their article (what does this mean?). If published, this will include your full peer review and any attached files.

**Do you want your identity to be public for this peer review?** For information about this choice, including consent withdrawal, please see our Privacy Policy.

Reviewer #1: **Yes: **Noyal Mariya Joseph

Reviewer #2: **Yes: **S Siva Kumar

---

## [Editor Report · Decision Letter 1]

8 Aug 2024

Perceived ease of use of Truenat among laboratory staff as a new diagnostic tool for TB in Nigeria: result of a pilot roll-out

PGPH-D-24-01529R1

Dear Dr. Edmund

We are pleased to inform you that your manuscript 'Perceived ease of use of Truenat among laboratory staff as a new diagnostic tool for TB in Nigeria: result of a pilot roll-out' has been provisionally accepted for publication in PLOS Global Public Health.

Best regards,

Priya Rajendran, PhD

Academic Editor